# Comparative Perceptual, Affective, and Cardiovascular Responses between Resistance Exercise with and without Blood Flow Restriction in Older Adults

**DOI:** 10.3390/ijerph192316000

**Published:** 2022-11-30

**Authors:** Thomas Parkington, Thomas Maden-Wilkinson, Markos Klonizakis, David Broom

**Affiliations:** 1Physical Activity, Wellness and Public Health Research Group, Department of Sport and Physical Activity, Sheffield Hallam University, Sheffield S1 1WB, UK; 2Lifestyle, Exercise and Nutrition Improvement Research Group, Department of Nursing and Midwifery, Sheffield Hallam University, Sheffield S1 1WB, UK; 3Centre for Sport, Exercise and Life Sciences, Coventry University, Coventry CV1 5FB, UK

**Keywords:** strength training, occlusion training, pain, acceptability, tolerability, adherence

## Abstract

Older adults and patients with chronic disease presenting with muscle weakness or musculoskeletal disorders may benefit from low-load resistance exercise (LLRE) with blood flow restriction (BFR). LLRE-BFR has been shown to increase muscle size, strength, and endurance comparable to traditional resistance exercise but without the use of heavy loads. However, potential negative effects from LLRE-BFR present as a barrier to participation and limit its wider use. This study examined the perceptual, affective, and cardiovascular responses to a bout of LLRE-BFR and compared the responses to LLRE and moderate-load resistance exercise (MLRE). Twenty older adults (64.3 ± 4.2 years) performed LLRE-BFR, LLRE and MLRE consisting of 4 sets of leg press and knee extension, in a randomised crossover design. LLRE-BFR was more demanding than LLRE and MLRE through increased pain (*p* ≤ 0.024, *d* = 0.8–1.4) and reduced affect (*p* ≤ 0.048, *d* = −0.5–−0.9). Despite this, LLRE-BFR was enjoyed and promoted a positive affective response (*p* ≤ 0.035, *d* = 0.5–0.9) following exercise comparable to MLRE. This study supports the use of LLRE-BFR for older adults and encourages future research to examine the safety, acceptability, and efficacy of LLRE-BFR in patients with chronic disease.

## 1. Introduction

Muscle weakness, presented as low muscle mass and strength, is strongly related to functional limitations, physical disability and mortality in older adults and patients with chronic disease [1,2]. Strong evidence has demonstrated that resistance exercise is a powerful intervention to develop muscle mass and strength which leads to an improvement in functional capacity, mobility, independence, chronic disease management, psychological well-being, and quality of life [3,4,5]. Moderate- to high-loads (≥60% of one repetition maximum; 1RM) are typically recommended for effective resistance exercise programming to induce hypertrophy and strength adaptations [6]. However, for some older adults and patients with chronic disease, such loads may not be safely performed due to underlying muscle weakness or musculoskeletal and/or cardiovascular disorders [3]. Therefore, alternative resistance exercise modes that use low loads and deliver comparable muscular adaptations to traditional resistance exercise are warranted and important.

Accordingly, many studies have advocated low-load resistance exercise (LLRE) with blood flow restriction (BFR) as an alternative to traditional resistance exercise [7,8]. The LLRE-BFR technique uses pneumatic cuffs to apply an external pressure around the proximal region of the exercising limb to partially restrict arterial blood flow and occlude venous return whilst lifting low loads (20–40% 1RM; 15–30 repetitions per set) [9]. During LLRE-BFR, the external pressure impairs intramuscular oxygen delivery and venous clearance of metabolites which elevates metabolic stress during sustained mechanical tension [10]. This increases the training response of systemic hormone production [11], myofibrillar and mitochondrial protein synthesis [12,13] and angiogenesis [14]. As a result, muscle size, strength and muscular endurance are substantially enhanced compared to LLRE without BFR [15], with incurred muscular improvements similar to that induced by HLRE [7,16].

The advantage of LLRE-BFR to elicit beneficial muscular adaptations through using low loads thereby imposing minimal mechanical stress on the musculoskeletal system could prove valuable to many older adults and patients with chronic disease [17,18,19]. However, LLRE-BFR has been shown to exacerbate the perceptual, affective, and cardiovascular response. Previous studies have reported increased ratings of perceived exertion (RPE) and pain, and reduced affect, task motivation and enjoyment to LLRE-BFR compared to LLRE without BFR [20,21], indicating a negative perceptual and affective response that could discourage exercise participation [22,23]. Additionally, LLRE-BFR increases heart rate (HR), systolic blood pressure (SBP), diastolic blood pressure (DBP), mean arterial pressure (MAP) and rate-pressure product (RPP) compared to LLRE without BFR [24,25]. This indicates an increased cardiovascular demand to LLRE-BFR which has prompted caution if prescribing LLRE-BFR to patients with chronic diseases [26].

These previous findings of potentially negative perceptual, affective, and cardiovascular responses to LLRE-BRE present as a barrier to exercise participation and limit its wider use by older adults and patients with chronic disease. However, to the best of our knowledge, no study has reported affective responses, and only two studies have reported perceptual responses (including pain) to LLRE-BFR specific to older adults [11,27]. Additionally, the extent the perceptual, affective, and cardiovascular responses to LLRE-BFR differ from traditional resistance exercise is unclear in the literature with studies reporting either lower, similar, or higher responses to LLRE-BFR compared with HLRE with inconsistencies in reports likely due to varying exercise protocols and BFR methods used [11,24,25,28,29,30,31,32,33]. Clarification is necessary if LLRE-BFR is to be presented as a viable alternative to traditional resistance exercise.

If the wider application of LLRE-BFR is to be used with older adults and patients with chronic disease, it is important to first explore the perceptual, affective, and cardiovascular responses to LLRE-BFR to determine the suitability of the protocol for older adults without muscle weakness or musculoskeletal and cardiovascular disorders. It is important that the LLRE-BFR protocol includes two different lower-body resistance exercises to represent an exercise session that would be used in practice. Additionally, it is essential to determine how LLRE-BFR compares to traditional moderate-load resistance exercise (MLRE) as this is the recommended prescription for older adults without resistance exercise experience [6]. Therefore, the aim of this study was to determine the impact of LLRE-BFR on perceptual, affective, and cardiovascular responses during two different resistance exercises within an exercise session and compare these responses to LLRE and MLRE in older adults.

## 2. Materials and Methods

### 2.1. Subjects

Twenty (10 males and 10 females) apparently healthy (i.e., free from disease) community-dwelling older adults were recruited to participate in the study from the local community via a promotion stall (located in Graves Health Centre, Sheffield, UK), social groups on Facebook and by word of mouth. Participants were required to meet the following inclusion criteria; (1) aged ≥60 years old; (2) were not engaged in resistance exercise in the previous six months; (3) did not self-report uncontrolled hypertension (>150/90 mmHg), musculoskeletal, neurological, or vascular disease/injury; (4) non-smokers, defined as not used tobacco and related products in the previous 6 months; and (5) did not meet more than one risk factor for thromboembolism, which includes the following; obesity (BMI > 30 kg/m^2^); diagnosed with Crohn’s disease; a past fracture of hip, pelvis, or femur; major surgery within the last 6 months; varicose veins; a family or personal history of deep vein thrombosis or pulmonary embolism [34]. Baseline participant characteristics are listed in Table 1. The sample size was utilised for the study to be sufficiently powered (α = 0.05, β = 0.80, medium effect 0.5; G*Power, version 3.1.9.3, Dusseldorf, Germany). All research procedures were approved by the Ethics Committee of Sheffield Hallam University (ER10932988) and conformed to the standards set by the Declaration of Helsinki. Prior to participation in the study, each participant was informed of the experimental procedures and risks that were associated with the study before giving written informed consent. Each participant was instructed not to participate in vigorous exercise 48 h prior to each exercise session. Additionally, each participant was instructed not to consume food or caffeine 2 h prior to each exercise session. If any muscle soreness in the legs was present on the day of an exercise session, then the visit was rescheduled for that participant so as not to influence the perceptual, affective, or cardiovascular response to exercise.

### 2.2. Methods

Participants attended the laboratory for two preliminary visits before the main exercise sessions. During the first preliminary visit, anthropometric data (stature, body mass, BMI, fat mass, muscle mass and waist circumference) and resting physiological data (lipid profile, blood glucose, SBP, DBP, MAP, PP, ankle-brachial pressure index; ABPI, and arterial occlusion pressure; AOP) were collected. Participants were then familiarised with exercising on the horizontal leg press machine (Pro 2 Seated Leg Press, Life-fitness, Chicago, IL, USA) and knee extension machine (SP100, TECA Fitness, Montesilvano, Italy) to become orientated to the exercise equipment. During the second preliminary visit, participants completed 1RM testing for both leg press and knee extension using the repetitions to failure method. This was adopted as it was anticipated participants would feel uncomfortable performing their true 1RM which may impact on participation and minimise the risk of injury. Participants were then familiarised with the three experimental conditions, which consisted of completing the first two sets of each protocol including experiencing the perceptual, affective, and cardiovascular measures to become orientated to the exercise protocols, study procedures and measures. Following preliminary visits, participants attended the laboratory a further three times to complete the experimental exercise sessions. A randomised crossover design was used to compare the perceptual, affective, and cardiovascular responses between (1) low-load resistance exercise with BFR (LLRE-BFR), (2) low-load resistance exercise (LLRE), and (3) moderate-load resistance exercise (MLRE). Each visit occurred at the same time of day and was separated by a minimum of 5 days to remove the effects of the previous visit.

#### 2.2.1. Predicted One Repetition Maximum

1RM for leg press and knee extension was predicted using the repetitions to failure method based on previously tested protocols to determine the load used for each experimental condition [35,36]. Following a standardised warm-up of 5 min light cycling, participants performed 10 repetitions at a load of low effort. The load was progressively increased until momentary failure occurred within 10 repetitions. Momentary failure was determined when, despite maximum effort, the participant was unable to complete a repetition through the full range of motion. 1RM was then predicted using the Brzycki equation [37]: load ÷ (1.0278 − [0.0278 × number of repetitions]). The Brzycki equation has shown excellent predictive accuracy of actual 1RM for leg press (0.96 ICC) and knee extension (0.99 ICC) [38]. The predicted 1RM for leg press and knee extension was 190.0 ± 68.6 kg and 64.0 ± 23.7 kg, respectively.

#### 2.2.2. Determination of Arterial Occlusion Pressure

Arterial occlusion pressure (AOP) was measured to determine individualised restrictive cuff pressure for participants during LLRE-BFR. In accordance with established methods [39], participants laid in a recumbent position in a quiet unlit room for 10 min. A 13 × 85 cm nylon cuff (SC12, Hokanson, Indianapolis, IN, USA) was applied at the most proximal portion of the thigh and an 8 MHz vascular Doppler probe (HI-Dop vascular Doppler, Ana Wiz, Surrey, UK) positioned on the posterior portion of the medial malleolus on the branches of the tibial artery of the same leg. The cuff was inflated (E20 Rapid cuff inflator and AG101 Cuff Inflator Air Source, Hokanson, Indianapolis, IN, USA) until interruption of the auditory signal of arterial blood flow suggesting arterial occlusion and the final pressure was recorded. This was then immediately repeated for the opposite leg. The mean AOP of both legs was used for the restrictive cuff pressure. Total AOP was 188.3 ± 24.8 mmHg.

#### 2.2.3. Resistance Exercise Protocols

Exercise sessions began with a standardised warm-up of 5 min light cycling. Participants then completed programmed protocols for seated 45° horizontal leg press then knee extension. Exercises were performed bilaterally with repetitions executed every 3 s (1.5 s during the concentric phase and 1.5 s during the eccentric phase) with support from a metronome. Exercises were separated by a 5 min passive rest period. LLRE-BFR and LLRE protocols involved 1 set of 30 repetitions followed by 3 sets of 15 repetitions for leg press and 4 sets of 15 repetitions for knee extension. Both exercises had 30 s rest periods between sets and were performed at a load of 20% 1RM. During LLRE-BFR, a 13 cm wide nylon pneumatic cuff (SC12L segmental pressure Cuff, Hokanson, Indianapolis, IN, USA) was placed around the proximal region of the legs. The cuff was inflated to 50% of AOP 15 s before starting either exercise. The pressure was maintained during the exercise bout and was deflated after the last repetition once perceptual and cardiovascular measures were obtained. To note, the cuff was deflated during the 5 min passive rest period between exercises. The MLRE protocol involved 4 sets of 10 repetitions with 60 s rest periods between sets and was performed at a load of 60% 1RM for both leg press and knee extension.

#### 2.2.4. Perceptual Responses

RPE was measured using the CR-20 Borg scale and pain using the CR-10^+^ modified pain scale immediately (within 5 s) following each set of exercise. Both CR-20 and CR-10^+^ scales have been shown to be valid and reliable in exercise and pain studies [40,41] and have been used to quantify RPE and pain in previous LLRE-BFR-related studies [21,32]. Session-RPE was measured using the CR-10 Borg scale 15 min following the completion of both exercises to indicate the perceived difficulty of the entire exercise session. Muscle soreness of the lower body was obtained 24 h and 48 h following each exercise session using the CR-10^+^ modified pain scale after requesting a rating via text message. To gauge soreness of the lower body, participants were asked to flex and extend both knees and press into their quadriceps with their hands before providing their ratings.

#### 2.2.5. Affective Responses

Affect (pleasure/displeasure) was measured using the Feeling Scale [42] immediately (within 5 s) following each set of exercise. The Feeling Scale is an 11-point scale ranging from very bad (−5) to very good (+5). The Feeling Scale has demonstrated face, content, and construct validity [42]. A modified Physical Activity Enjoyment Scale (PACES) [43] was used to indicate levels of enjoyment of the entire exercise session measured 15 min following the completion of exercise. The PACES included 5 items (enjoy, like, fun, physical feeling, and frustration) displaying two contrasting statements about exercise (e.g., “I like it” and “I dislike it”). Between the two statements, participants rated their agreement with each statement on a 7-point Likert-type scale. The Physical Activity Affect Scale (PAAS) [44] was used to assess the affective response to the exercise session and was measured upon the arrival to the laboratory at rest and following 15 min after the completion of exercise. The PAAS questionnaire includes 12 feelings which are equally divided into 4 subscales: positive affect (enthusiastic, energetic, and upbeat), negative affect (miserable, discouraged, and crummy), tranquillity (calm, relaxed, and peaceful) and fatigue (fatigued, tired, and worn-out). Participants were asked to rate their current affective state for each item on a scale; do not feel (0), feel slightly (1), feel moderately (2), feel strongly (3) or feel very strongly (4). A mean score for each subscale was calculated and used for analysis. The PAAS is sensitive to affective changes during exercise [45], and has shown convergent and discriminant validity in both active and sedentary individuals [46]. Three separate visual analogue scales (VAS) were used to indicate enjoyment, fatigue and perceived effectiveness of the entire exercise session measured 15 min following the completion of all exercise. All VAS spanned a single 10 cm horizontal line with a headline statement at the top. To the extreme left of the line was an answer that indicated no agreement with the headline statement (e.g., no enjoyment/no fatigue/not at all effective) and to the extreme right the statement indicated strong agreement (e.g., very enjoyable/very fatiguing/very effective).

#### 2.2.6. Cardiovascular Responses

Blood pressure was assessed using an automatic monitor (HEM-8712, Omron, Healthcare, Kyoto, Japan) immediately following the last set of each exercise according to standardised operating procedures. HR was monitored using a traditional chest strap (TICKR, Wahoo, Atlanta, GA, USA) throughout the exercise session and recorded every 5 s excluding rest periods. These data were used to calculate HR_mean_, HR_peak_, MAP (calculated as 1/3 (SBP − DBP) + DBP), RPP (calculated as SBP × HR/100) and PP (calculated as SBP − DBP).

#### 2.2.7. Statistical Analysis

Data are presented as means ± SEM unless indicated otherwise. Prior to analysis, the Shapiro–Wilk test confirmed that data was normally distributed. Following this, linear mixed models were performed on all data. RPE, pain, and affect were compared between conditions (LLRE-BFR, LLRE and MLRE) and time (e.g., between each set within an exercise), with condition and time set as fixed factors and participants set as a random factor. Separate analyses were performed for leg press and knee extension. The mean RPE, pain and affect across sets 1–4 were compared between conditions and exercise (leg press and knee extension), with condition and exercise set as fixed factors and participants set as a random factor. Muscle soreness and PAAS were compared between conditions and time with condition and time set as fixed factors and participants set as a random factor. Session-RPE, PACES and VAS were compared between conditions with condition set as a fixed factor and participants set as a random factor. Where a significant interaction or main effect was observed Bonferroni post hoc assessment was used to identify where the differences occurred. Magnitude of differences was determined using Cohen’s *d* (difference in the mean divided by the standard deviation of the difference; small effect = 0.20–0.49, moderate effect = 0.50–0.79, and large effect = ≥0.80). Statistical analysis was conducted using SPSS (Version 26, Chicago, IL, USA), with statistical significance set at *p* ≤ 0.05.

## 3. Results

### 3.1. Adverse Events

One adverse event occurred outside of the testing environment with a participant who presented with superficial thrombophlebitis in their right leg 3 weeks after completing all study commitments. The participant made a full recovery following four weeks of treatment. In consultation with the research teams’ clinical expert, it was deemed this adverse event was not caused due to the exercise protocols completed in this study primarily because of the length of time in which the issue presented.

### 3.2. Total Repetitons Completed

A high level of completion was observed in this study with only two participants unable to complete all programmed repetitions for LLRE-BFR during sets 3 and 4 of knee extension due to fatigue.

### 3.3. RPE, Pain, and Affect

A significant condition by time interaction (*p* = 0.005) was observed for RPE during leg press (Figure 1). RPE increased from pre to set 1 (*p* < 0.001, *d* = 1.6–2.7) then remained constant until set 4 for all conditions. RPE was lower for every set during LLRE compared with MLRE (*p* < 0.001, *d* = −1.1), while set 4 was lower during LLRE compared with LLRE-BFR (*p* = 0.033, *d* = −0.8). Similarly, a significant condition by time interaction (*p* < 0.001) was observed for RPE during knee extension (Figure 1). RPE increased from pre to set 1 (*p* < 0.001, *d* = 2.0–3.2) then increased across sets 1–4 (*p* ≤ 0.019, *d* = 0.8–2.1) for all conditions. RPE was lower for sets 1–3 during LLRE compared with MLRE (*p* ≤ 0.008, *d* = −0.9–−1.2), while sets 2–4 were lower during LLRE compared with LLRE-BFR (*p* ≤ 0.002, *d* = −0.9–−1.0). Differences between LLRE-BFR and MLRE occurred only at set 1 with RPE lower during LLRE-BFR (*p* = 0.004, *d* = −0.9). For comparisons of mean RPE, significant condition (*p* < 0.001) and exercise (*p* < 0.001) main effects were observed (Figure 2). Post hoc analyses confirmed mean RPE was lower for LLRE compared to LLRE-BFR and MLRE (*p* < 0.001, *d* = −1.1–−1.6) and mean RPE for knee extension was higher than leg press (*p* < 0.001, *d* = 1.4) for all conditions.

A significant condition by time interaction (*p* = 0.002) was observed for pain during leg press (Figure 1). Pain increased from pre to set 1 (*p* ≤ 0.001, *d* = 1.0–1.5) for all conditions then increased from set 1–set 4 for LLRE-BFR (*p* < 0.001, *d* = 1.1) but remained unchanged for LLRE and MLRE. Pain was higher during LLRE-BFR for every set compared with LLRE (*p* ≤ 0.024, *d* = 0.8–1.4) and sets 3–4 compared with MLRE (*p* ≤ 0.005, *d* = 0.9–1.0). Likewise, a significant condition by time interaction (*p* < 0.001) was observed for pain during knee extension (Figure 1). Pain increased from pre to set 1 (*p* ≤ 0.018, *d* = 0.8–1.2) then increased across sets 1–4 (*p* ≤ 0.033, *d* = 0.8–2.0) for all conditions. Pain was similar between conditions at pre and sets 1–2. At sets 3–4, pain was higher for LLRE-BFR compared to LLRE and MLRE (*p* ≤ 0.001, *d* = 1.0–1.2). For comparisons of mean pain, significant condition (*p* < 0.001) and exercise (*p* < 0.001) main effects were observed (Figure 2). Post hoc analyses confirmed mean pain was higher for LLRE-BFR compared to LLRE and MLRE (*p* < 0.001, *d* = 1.0–1.4) and mean pain was higher for knee extension than leg press (*p* < 0.001, *d* = 1.7) for all conditions.

A significant condition main effect (*p* = 0.031) was observed for affect during leg press with affect lower during LLRE-BFR compared to LLRE and MLRE (*p* = 0.048, *d* = 0.5) (Figure 1). For knee extension, significant condition (*p* < 0.001) and time (*p* < 0.001) main effects for affect were observed (Figure 1). Affect decreased from set 1–4 (*p* < 0.001, *d* = 1.5) for all conditions. Additionally, affect was lower for LLRE-BFR compared to LLRE (*p* < 0.001, *d* = −1.32) and MLRE (*p* < 0.001, *d* = −0.93). For comparisons of mean affect, significant condition (*p* = 0.003) and exercise (*p* < 0.001) main effects were observed (Figure 2). Post hoc analyses confirmed mean affect was lower for LLRE-BFR compared to LLRE and MLRE (*p* ≤ 0.047, *d* = −0.6–−0.8) and mean affect was lower for knee extension than leg press (*p* < 0.001, *d* = −1.0) for all conditions.

### 3.4. Session-RPE

A significant condition main effect (*p* < 0.001) was observed for session-RPE (Figure 3). The highest rating of session-RPE followed LLRE-BFR compared to LLRE (*p* < 0.001, *d* = 1.7) and MLRE (*p* = 0.002, *d* = 0.8) and session-RPE was higher following MLRE compared to LLRE (*p* = 0.001, *d* = 0.9).

### 3.5. Cardiovascular Responses

Cardiovascular responses are presented in Table 2. A significant condition main effect (*p* = 0.006) for HR_mean_ was observed with higher HR_mean_ during LLRE-BFR compared to LLRE and MLRE (*p* ≤ 0.020, *d* = 0.7). For HRpeak, a significant condition by exercise interaction (*p* = 0.011) was observed. Post hoc analysis confirmed HRpeak was higher for LLRE-BFR compared to LLRE and MLRE (*p* ≤ 0.031; *d* = 0.7–0.9) during knee extension. Additionally, HRpeak was greater during knee extension than leg press during LLRE-BFR (*p* = 0.023; *d* = 0.7). A significant main effect of exercise (*p* = 0.037) was observed for SBP with higher SBP during knee extension than leg press (*p* = 0.037; *d* = 0.5) for all conditions. Significant main effects of condition (*p* = 0.001) and exercise (*p* = 0.020) were observed for RPP. RPP was lowest during LLRE compared to LLRE-BFR and MLRE (*p* ≤ 0.032; *d* = −0.6–−0.9) and RPP was higher during knee extension than leg press (*p* = 0.020; *d* = 0.5) for all conditions. A significant main effect of exercise (*p* = 0.014) was observed for PP with higher values occurring during knee extension than leg press for all conditions (*p* = 0.014; *d* = 0.6).

### 3.6. Physical Activity Enjoyment Scale

A significant condition main effect (*p* = 0.004) was observed for PACES with the sum of PACES lower following LLRE-BFR compared to LLRE (*p* = 0.033, *d* = −0.6) (Figure 4).

### 3.7. Physical Activity Affect Scale

Significant time main effects were observed for PAAS subscales positive affect (*p* = 0.006), negative affect (*p* = 0.035) and tranquillity (*p* < 0.001) (Table 3). Post hoc analyses confirmed positive affect increased from pre to post (*p* = 0.006, *d* = 0.6), negative affect decreased from pre to post (*p* = 0.035, *d* = −0.5) and tranquillity increased from pre to post (*p* < 0.001, *d* = 0.9) for all conditions.

### 3.8. Visual Analogue Scales

Enjoyment and perceived effectiveness were high, and fatigue was mild following all conditions with no significant condition main effect observed (Figure 5).

### 3.9. Muscle Soreness

Participants had no leg muscle soreness at baseline. Muscle soreness at 24 h (LLRE-BFR = 0.7 ± 0.2 CR-10, LLRE = 0.3 ± 0.1 CR-10 and MLRE 1.0 ± 0.3) and 48 h (LLRE-BFR = 0.4 ± 0.2 CR-10, LLRE = 0.0 ± 0.0 CR-10 and MLRE 0.4 ± 0.2) was low for all conditions. Statistically significant condition (*p* = 0.002) and time (*p* < 0.001) main effects were observed for muscle soreness. Muscle soreness tended to increase from pre to 24 h (*p* < 0.001, *d* = 1.4) and 48 h (*p* = 0.031, *d* = 0.8) following all conditions. Additionally, muscle soreness was greater following both LLRE-BFR and MLRE compared to LLRE (*p* ≤ 0.049, *d* = 0.5–0.8).

## 4. Discussion

This investigation explored the perceptual, affective, and cardiovascular responses to a bout of LLRE-BFR involving two resistance exercises and compared the responses to a bout of LLRE and MLRE in older adults. The main findings of the study indicate that LLRE-BFR was more demanding than LLRE and MLRE, predominately through increased pain and reduced affect, though was enjoyed and promoted a positive affective response post exercise comparable to MLRE. Additionally, knee extension increased effort and pain, and reduced affect compared to leg press for all conditions, which may be worsened by BFR. Our findings provide new insights into the acceptability of LLRE-BFR and have practical relevance to practitioners considering implementing LLRE-BFR as an alternative to traditional resistance exercise.

An increase in pain during LLRE with the addition of BFR is extremely consistent within the literature indicating the additional stress of BFR [47]. Our findings of “moderate” to “strong” pain resulting from LLRE-BFR are comparable to other studies using similar LLRE-BFR protocols that recruited older [27] and younger [32] adults. Pain resulting from LLRE-BFR increased following each set of exercise which exceeded MLRE and is likely comparable to HLRE [32,48]. The increased pain is a result of the BFR stimulus accelerating the development of peripheral fatigue primarily through the reduction of venous clearance of fatigue-related metabolites, evidenced by increased phosphocreatine depletion, reduced muscle pH, augmented blood lactate and impaired muscle contractile function [25,49,50]. Metabolic stress is a key mechanism contributing to the beneficial muscular adaptations from LLRE-BFR training [10]; therefore, some level of muscular pain should be expected when performing LLRE-BFR. Though elevated pain is associated with reduced affect and task motivation during exercise [20] which can deter future LLRE-BFR participation [22].

To manage the pain, practitioners must first consider the restrictive cuff pressure and the exercise load as these factors have a substantial impact on the pain experienced [21,48]. In the present study a restrictive cuff pressure of 50% AOP and exercise load of 20% 1RM was used. To the authors’ knowledge, this protocol is the lowest effective restrictive cuff pressure and exercise load applied to older adults reported in the literature [51]. Higher restrictive cuff pressures (80% AOP) and exercise loads (40% 1RM) are associated with “very strong” pain with some individuals reporting near maximal ratings [21,48]. Another consideration for practitioners should be exercise volume. Given there is a step increase in pain per exercise set, the initial lower volume in the LLRE-BFR programme may reduce pain experienced and increase the likelihood of individuals completing all programmed repetitions. Such a strategy may boost exercise self-efficacy and encourage programme adherence [52]. Exercise volume can progressively increase through the programme to the target 30-15-15-15 repetitions as per guidelines [9] as the adaptive response to training provides a greater tolerance to the stimulus [53].

Enjoyment and affective response are posited as important factors determining future exercise participation [23] and infer the acceptability of an exercise modality [54]. Individuals who experience enjoyment, improvements in affect, and low fatigue from exercise report more positive attitudes, exercise self-efficacy and intentions to exercise three months later [45,55]. It has been previously reported the addition of BFR to LLRE [20] and to walking [56] reduces the enjoyment response from exercise. Additionally, mood status assessed via the Brunel Mood Scale and Profile of Mood States was negatively affected by LLRE-BFR predominately through reduced vigour and increased fatigue immediately post exercise [20,33]. Furthermore, tranquillity and physical exhaustion assessed by the Exercise Induced Feelings Inventory has shown to decrease and increase respectively immediately following LLRE-BFR but return to baseline within 15 min [57]. Our findings of a comparable enjoyment and affective response following LLRE-BFR and MLRE should encourage practitioners to use LLRE-BFR with older adults as an alternative to traditional resistance exercise.

The present study demonstrated higher cardiovascular responses with LLRE-BFR compared to LLRE and MLRE. Previous studies have consistently shown the addition of BFR to LLRE augments HR and blood pressure responses [58]. This may be contributed by the reduced blood flow during muscular exertion that augments metabolic accumulation and pain and influences the pattern of muscle recruitment which increases the exercise pressor reflex resulting in an enhanced autonomic cardiovascular response [59]. Similar HR and blood pressure responses to the present study have been previously observed [24,25]. It is important to note that these blood pressure measures were taken immediately following the conclusion of the exercise set and blood pressure is likely higher when measured during muscular contraction [25].

While some studies report the cardiovascular responses to LLRE-BFR exceed HLRE [24,60], others report comparable or lower responses [25,28]. Cardiovascular responses reported in the BFR literature are highly variable, likely due to differing measurement techniques, exercise protocols and populations used between studies. Importantly, the observed increases in HR and blood pressure are consistently within the normal range expected during exercise [61] and to date, no cardiovascular events during LLRE-BFR have been reported in the literature from research or in the field [62,63]. Given the number of studies using LLRE-BFR in older adults, LLRE-BFR is likely safe for this group. However, more research examining the safety and cardiovascular responses to LLRE-BFR in patients with chronic diseases is required prior to the wider use of the technique for these special populations.

LLRE-BFR methodology is critical to the experience of the technique. Restrictive cuff pressures [48,64], cuff width [65,66], type of cuff [67] and exercise load [21] have been shown to substantially effect the perceptual and cardiovascular response. An interesting finding in the present study was an increase in perceptual and cardiovascular response during knee extension compared to leg press for all conditions, which may be worsened with BFR. This disagrees with findings from Scott et al. [24] who reported an increase in RPE, HR, SBP, DBP, MAP and RPP with leg press than knee extension. Conflicting observations may be due to the different exercise protocols between studies, with Scott et al., [24] employing a lower exercise volume of 1 set of 20 repetitions followed by 2 sets of 15 repetitions for both leg press and knee extension. However, studies which have employed the same exercise protocols as the present study have observed trends of increased RPE and pain to knee extension compared to leg press, but this was not statistically analysed [32,67].

The discrepancy in perceptual and cardiovascular response between leg press and knee extension could be due to physiological changes caused by mechanical differences. More volume of exercise can be completed with large muscle group exercises (e.g., leg press) compared with smaller muscle group exercises (e.g., knee extension) at the same relative intensity [68,69]. This may be due to asynchronous motor unit recruitment during submaximal exercise which serves to delay fatigue [68]. Thereby, individuals may tolerate leg press to a greater degree than knee extension at a set exercise load when a high exercise volume is performed. Why BFR may exaggerate the increase in perceptual and cardiovascular response to knee extension compared to leg press is unclear. Although speculative, the dynamic hip extension-flexion action during leg press may facilitate venous flow with BFR which could limit blood pooling and metabolic accumulation proximal to the restrictive cuff compared to static hip action during knee extension. Furthermore, the contribution of hip extensor muscles during leg press may not be significantly affected by the BFR stimulus and thus less susceptible to the accumulation of metabolic stress associated with BFR. Lower body resistance exercises leg press and knee extension are most prescribed with LLRE-BFR [62]. Determining exercise that is perceived favourably is useful for practitioners using LLRE-BFR in research or in rehabilitation therapy to develop protocols that encourage exercise adherence.

The study is not without limitations. This study was acute in nature therefore only inferences can be made about the long-term suitability of LLRE-BFR for older adults. We recognise that although participants completed the conditions in a randomised order, we cannot dismiss a repeated-bout effect that could contribute to the altered perceptual response to resistance exercise. RPE and pain have been shown to subside after repeated sessions of resistance exercise, suggesting an adaptive effect to psychological markers (e.g., sense of effort and pain) that facilitates greater tolerance [53,70]. Additionally, our interpretation of the perceptual and cardiovascular differences between leg press and knee extension may have been influenced by an order effect as all participants first undertook leg press then knee extension. To substantiate our findings future studies should randomise the delivery of exercises performed. We acknowledge that the participants recruited for this study were all physically active and interested in exercise. Older adults and patients with chronic disease with muscle weakness or musculoskeletal and/or cardiovascular disorders may respond to the exercise conditions differently from those included in the present study.

## 5. Conclusions

Overall, the present study demonstrated an exercise session of LLRE-BFR was more demanding than LLRE and MLRE, predominately through increased pain and reduced affect. Interestingly, knee extension incurred greater perceptual and cardiovascular responses than leg press during the exercise sessions, which may be worsened by BFR. Potential negative effects of pain and affect from LLRE-BFR during exercise did not impact the enjoyment and affective response post exercise. LLRE-BFR was enjoyed and promoted a positive affective response comparable to MLRE. These factors are important in predicting future exercise engagement. Our findings provide new insights into the acceptability of LLRE-BFR and have practical relevance to practitioners considering implementing LLRE-BFR as an alternative to traditional resistance exercise. This study supports the use of LLRE-BFR for older adults and encourages future research to examine the safety, acceptability, and efficacy of LLRE-BFR in patients with chronic disease.

## Figures and Tables

**Figure 1 ijerph-19-16000-f001:**
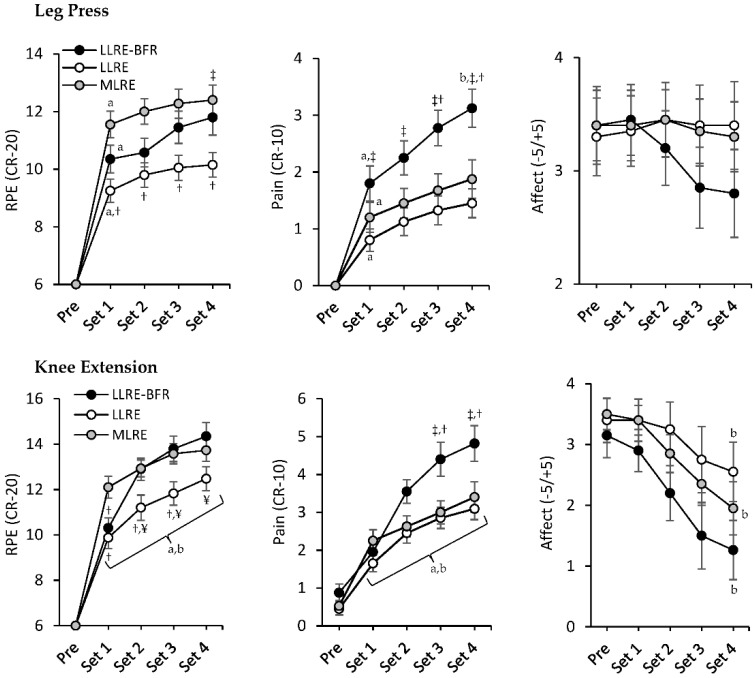
RPE, pain and affect for each exercise and immediately following each set. ^a^ Significant difference between pre and set 1; ^b^ significant difference between set 1 and set 4; ^¥^ significantly different to LLRE-BFR at same time point; ^‡^ significantly different to LLRE at same time point; ^†^ significantly different to MLRE at same time point. Brackets indicate differences for all three conditions.

**Figure 2 ijerph-19-16000-f002:**
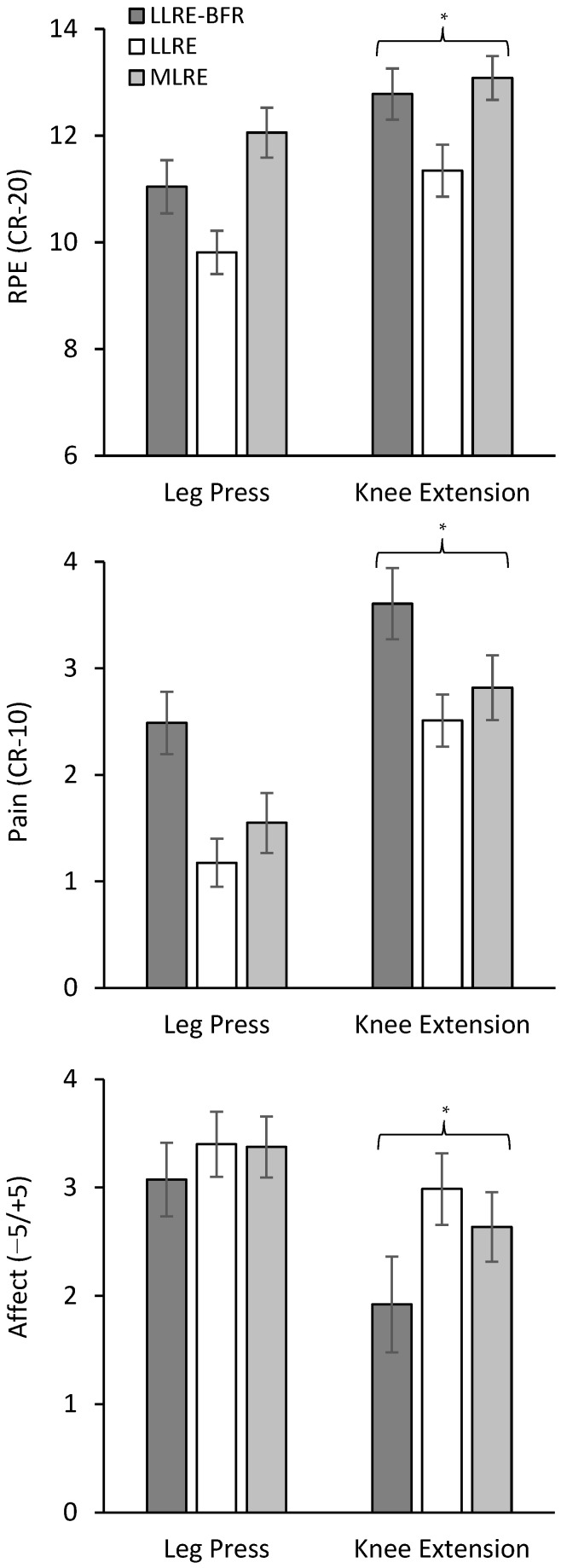
Mean RPE, pain and affect across sets 1–4 to display differences between conditions and exercises. * Significantly different to leg press. Brackets indicate differences.

**Figure 3 ijerph-19-16000-f003:**
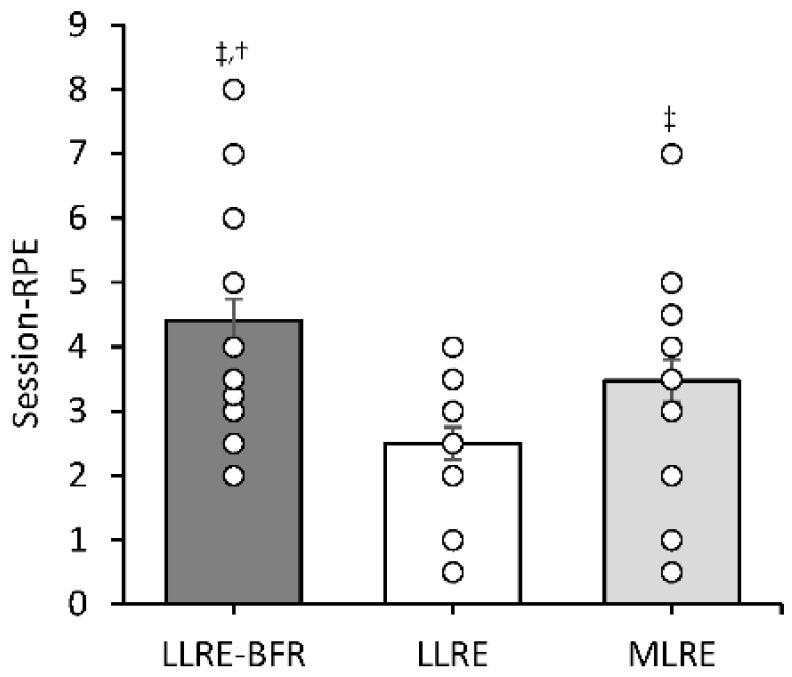
Session-RPE recorded 15 min post exercise session. Individual data points are displayed to provide information regarding individual responses to exercise. ^‡^ Significantly different to LLRE; ^†^ significantly different to MLRE.

**Figure 4 ijerph-19-16000-f004:**
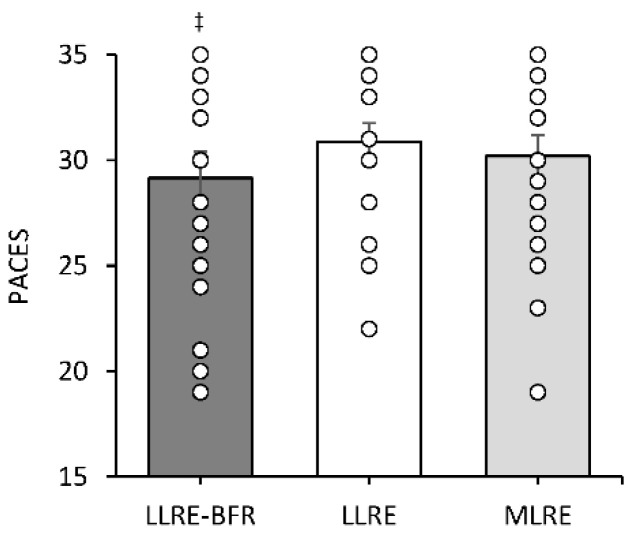
Sum of Physical Activity Enjoyment Scale measured 15 min post exercise session displaying individual data points to provide information regarding individual responses to exercise. ^‡^ Significantly different to LLRE.

**Figure 5 ijerph-19-16000-f005:**
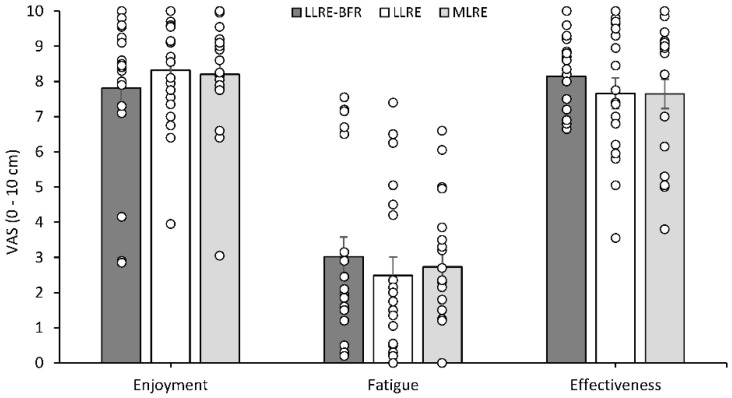
Visual analogue scales of enjoyment, fatigue, and perceived effectiveness recorded 15 min post exercise session. Individual data are displayed to provide information regarding individual responses to exercise.

**Table 1 ijerph-19-16000-t001:** Participants baseline characteristics.

Variable	Total (*n* = 20)	Males (*n* = 10)	Females (*n* = 10)
Age (years)	64.3 ± 4.2	63.6 ± 3.2	64.9 ± 5.2
Stature (cm)	171.3 ± 9.8	179.0 ± 4.9	163.7 ± 6.8
Body mass (kg)	75.1 ± 11.5	81.2 ± 11.5	68.9 ± 7.9
BMI (kg/m^2^)	25.6 ± 3.7	25.3 ± 3.4	25.9 ± 4.1
Fat mass (%)	24.8 ± 10.6	21.6 ± 8.0	28.0 ± 12.3
Muscle mass (%)	39.1 ± 10.6	44.0 ± 4.6	34.2 ± 12.8
Waist circumference (cm)	83.2 ± 8.7	86.2 ± 9.6	79.9 ± 6.6
Lipid profile			
Total	5.4 ± 0.8	5.1 ± 0.5	5.7 ± 1.0
LDL	3.1 ± 0.8	2.8 ± 0.6	3.6 ± 0.9
HDL	1.5 ± 0.4	1.5 ± 0.5	1.6 ± 0.4
TRI	1.5 ± 0.9	1.8 ± 1.0	1.2 ± 0.5
Blood glucose (mmol/L)	5.4 ± 0.8	5.5 ± 0.7	5.3 ± 1.0
SBP (mmHg)	131.8 ± 8.0	129.4 ± 7.7	133.6 ± 7.8
DBP (mmHg)	84.0 ± 8.3	83.5 ± 9.5	84.5 ± 8.0
MAP (mmHg)	99.9 ± 7.7	98.8 ± 8.7	101.2 ± 6.7
PP (mmHg)	47.8 ± 6.2	45.9 ± 4.5	50.0 ± 7.4
ABPI	1.2 ± 0.1	1.2 ± 0.1	1.2 ± 0.1
IPAQ-SF			
Physical activity category	moderate	moderate	moderate

Note. BMI, body mass index; LDL, low density lipid profile; HDL, high density lipid profile; TRI, triglycerides; SBP, systolic blood pressure; DBP, diastolic blood pressure; MAP, mean arterial pressure; PP, pulse pressure; ABPI, ankle brachial pressure index; IPAQ-SF, international physical activity questionnaire-short form; values are presented as mean ± SD.

**Table 2 ijerph-19-16000-t002:** Cardiovascular responses to leg press and knee extension.

Leg Press	LLRE-BFR	LLRE	MLRE
HR_mean_	86 ± 3	84 ± 3	84 ± 3
HR_peak_	94 ± 3	93 ± 3	96 ± 3
SBP	145 ± 3	131 ± 4	140 ± 4
DBP	87 ± 2	83 ± 2	85 ± 3
MAP	106 ± 2	99 ± 3	103 ± 3
RPP	132 ± 6	117 ± 4	134 ± 7
PP	58 ± 3	48 ± 2	54 ± 3
Knee Extension			
HR_mean_	88 ± 3	83 ± 2	83 ± 2
HR_peak_	101 ± 3 ^‡†^*	93 ± 3	94 ± 3
SBP	144 ± 5	143 ± 4	146 ± 3
DBP	86 ± 2 *	84 ± 2 *	84 ± 2 *
MAP	106 ± 3	104 ± 2	105 ± 2
RPP	146 ± 6 *	131 ± 5 *	135 ± 5 *
PP	58 ± 4 *	53 ± 3 *	61 ± 3 *

Note. HR, heart rate; SBP, systolic blood press; DBP, diastolic blood pressure; MAP, mean arterial blood pressure; RPP, rate pressure product; PP, pulse pressure. ^‡^ Significantly different to LLRE; ^†^ significantly different to MLRE; * significantly different to leg press.

**Table 3 ijerph-19-16000-t003:** Physical Activity Affect Scale measured pre and post exercise session.

	LLRE-BFR		LLRE		MLRE	
PAAS (0–4)	Pre	Post	Pre	Post	Pre	Post
Positive affect	2.5 ± 0.3	2.9 ± 0.2 ^a^	2.8 ± 0.2	2.9 ± 0.2 ^a^	2.8 ± 0.2	3.0 ± 0.2 ^a^
Negative affect	0.2 ± 0.1	0.1 ± 0.1 ^a^	0.2 ± 0.1	0.1 ± 0.1 ^a^	0.3 ± 0.1	0.1 ± 0.1 ^a^
Fatigue	0.6 ± 0.1	0.8 ± 0.2	0.6 ± 0.1	0.5 ± 0.1	0.6 ± 0.1	0.7 ± 0.1
Tranquillity	2.6 ± 0.2	2.9 ± 0.2^a^	2.7 ± 0.2	2.8 ± 0.2 ^a^	2.6 ± 0.2	3.1 ± 0.2 ^a^

Note. ^a^ Significant difference between pre and post.

## Data Availability

Not applicable.

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
