# Peer review of "Comparative Perceptual, Affective, and Cardiovascular Responses between Resistance Exercise with and without Blood Flow Restriction in Older Adults"

_ijerph, 2022, doi:10.3390/ijerph192316000_

Round 1

Reviewer 1 Report

Thank you for the opportunity to review your manuscript, which reports a randomised crossover design study comparing low-load resistance exercise (LLRE) with blood flow restriction (BFR) to LLRE and moderate-load resistance exercise (MLRE). It was a pleasure to read a well-written study that followed an appropriately rigorous design and with clearly organised results for presentation. The findings in support of LLRE-BFR for older adults offer what looks likely to be an effective alternative for patients with chronic disease.

My suggestions are for minor corrections here and there throughout the manuscript.

Line 196 ‘The Feeling Scale is a 11-point…”. Should be ‘an 11-point…’

Line 203. Title case for PAAS?

Lines 217-219. Delete commas in the sentence

Lines 398-399 “Though the conflict with LLRE-BFR is elevated pain during exercise reduces affect and task motivation [20] and 399 can deter future participation [22].” Requires punctuation or rewording for clarity. Currently a garden path sentence.

Line 401 Requires a comma following pain: “To manage the pain practitioners must first consider…”

Line 421 “…via Brunel Mood Scale…” Insert ‘the’ following via & delete comma following States

Lines 459-460 “Though, studies who have employed similar exercise protocols to the present study, trends of increased RPE and pain to knee extension compared to leg press were observed [32, 67].” Needs rewording to maintain internal sentence flow (and rethink commas).

Lines 499-500 “…our findings provide new insights into the acceptability of LLRE-BFR and has practical…” Needs rewording to maintain internal sentence flow (verb mismatch).

Reviewer 2 Report

I want to commend the authors on their investigation into the cardiovascular, affective and perceptual outcomes of an acute session of BFR compared to moderate intensity exercise in older adults. I think the manuscript was well written, substantiative and will be additive to the body of literature on the topic. Here is some minor comments, but overall, I am very impressed with the work of the author group in communicating clearly the methodology and associated conclusions.

Ln 111: good!

Ln 164: was additional rest provided for the opposite side or was it taken immediately post-AOP on the first leg? Also – any ICC values for reliability of the assessor of AOP?

Ln 168: Was exercise order the same for all participants? This could have impacted the findings of leg press vs. leg extension on the observed outcomes.

Ln 192: What muscle? Quadriceps?

Ln 224: Was the cuff still inflated on the legs when BP was taken? Or was it deflated then BP was assessed?

Ln 323: That is fascinating that HRpeak was less in leg press than leg extensions

Ln 473-5: I would imagine this is the most likely explanation. More unoccluded muscle group recruitment.

On a side note:

I would recommend you check out this paper if you haven’t seen it in your literature search: https://www.frontiersin.org/articles/10.3389/fresc.2021.697082/full

I think you all would find it interesting given the topic.

Reviewer 3 Report

There are several main issues of this paper:

1.       The result presentation and interpretation are not clear. For instance, for cardiovascular response, what does it mean to higher systolic pressure in LLRE-BFR group as compared to other groups?  Too many abbreviations, what other abbreviations mean ? you need to give a dictionary under the table.

2.       It is hard to understand the implication of LLRE-BFR , is it bad or good?

3.       In Table 3,  the  Pre in three groups are different, why?  Carry over effect?  You need to present change between pre and post; otherwise, it is not clear what you mean
